

# Kaluza-Klein spectrometry for $\text{AdS}_3$ vacua

## Camille Eloy⋆

Univ Lyon, Ens de Lyon, Univ Claude Bernard, CNRS,
Laboratoire de Physique, F-69342 Lyon, France

⋆ camille.eloy@ens-lyon.fr

## Abstract

We use exceptional field theory to compute Kaluza-Klein mass spectra around $\text{AdS}_3$ vacua that sit in half-maximal gauged supergravity in three dimensions. The formalism applies to any vacuum that arises from a consistent truncation of higher-dimensional supergravity, no matter what symmetries are preserved. We illustrate its efficiency by computing the spectra of $\mathcal{N} = (2, 0)$ and $\mathcal{N} = (1, 1)$ six-dimensional supergravities on $\text{AdS}_3 \times S^3$ and of type II supergravity on $\text{AdS}_3 \times S^3 \times S^3 \times S^1$.

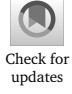

# 1 Introduction

The compactification of a higher-dimensional theory induces the appearance of infinitely many massive fields in the low-dimensional theory, which organize into multiplets of the symmetry group associated to the space of compactification. These massive excitations are often called Kaluza-Klein towers. They play an important role, as they can for example be used to predict the stability of the theory around a given vacuum, or to compute conformal dimensions of operators in the context of the AdS/CFT holographic correspondence. However, the computation of the Kaluza-Klein spectrum is in general involved. It demands the linearization of the higher-dimensional equations of motion, the expansion of all fields in harmonics of the internal space and to disentangle the resulting equations to deduce the mass matrices. This program has been successfully applied only for backgrounds that enjoy a coset space structure with a large isometry group (see Ref. [1–5] for examples), or for general backgrounds but restricting to the spin-2 fields [6].

Recently, a new technique that uses the framework of exceptional field theory [7–10] has been developed [11, 12]. Exceptional field theories provide a duality covariant formulation of higher dimensional supergravities, and in particular allow the construction of consistent truncations, *via* reduction ansätze of the higher-dimensional fields directly expressed in terms of the ones of the low-dimensional gauged supergravity. For a given low-dimensional background that arises from a consistent truncation, one can extend these ansätze so that they also describe the linearized higher-dimensional fluctuations around the background. This allows the computation of the mass matrices of the full Kaluza-Klein towers and, in particular, makes it possible to compute the spectrum around vacua with few or no remaining symmetries.

In Ref. [11, 12], this technique has been worked out for compactifications to five- and four-dimensional maximal supergravities. The purpose of this article is to extend these new tools to vacua that sit in half-maximal supergravity in three dimensions. The three dimensional case is distinguished: the on-shell duality between scalars and gauge fields together with the non-propagating nature of the gravitational multiplet provide a very rich structure, which admits a large amount of AdS$_3$ vacua. Half-maximal gauged supergravities in three dimensions have been constructed in Ref. [13,14] by deforming the half-maximal ungauged theory of Ref. [15]. They describe the couplings between the non-propagating $\mathcal{N} = 8$ supergravity multiplet to $p$ scalar multiplets, and feature a SO(8, $p$) global symmetry. Different gauge groups, embedded into SO(8, $p$), are realized using the embedding tensor formalism. The relevant framework is then the duality covariant SO(8, $p$) exceptional field theory of Ref. [16].

We compute here the expressions of the mass matrices for spin-2, vector and scalar fields around AdS$_3$ vacua of three-dimensional half-maximal supergravity. We then illustrate the efficiency of these new tools on several examples. We first test the formulas using $\mathcal{N} = (2, 0)$ six-dimensional supergravity on AdS$_3 \times S^3$. This example is particular, as its structure is sufficiently constrained by supersymmetry to allow a computation of the spectrum using only group theory [17]. We then turn to $\mathcal{N} = (1, 1)$ supergravity in six dimensions, where the same vacuum AdS$_3 \times S^3$ preserves only half of the supersymmetries, so that group theory fails to predict the masses in the spectrum. We finally consider ten-dimensional supergravity on AdS$_3 \times S^3 \times S^3 \times S^1$, which constitutes another example where representation theory is not sufficient and an explicit calculation is needed [18, 19].

The article is organized as follows. In Sec. 2, we review the framework of the SO(8, $p$) exceptional field theory. We then present in Sec. 3 how to generalize the compactification ansätze to include Kaluza-Klein fluctuations, and derive the expressions of the mass matrices of the bosonic fields. We illustrate in Sec. 4 the method on three examples, and finally summarize our findings and conclude in Sec. 5.

# 2  SO(8, $p$) exceptional field theory

First constructed in Ref. [16], the SO(8, $p$) exceptional field theory is a duality-covariant formulation of half-maximal supergravity, designed for compactifications to three spacetime dimensions[1]. Its fields live on a set of coordinates which contains three-dimensional external coordinates $\{x^\mu\}$, $\mu \in [\![1, 3]\!]$, and internal coordinates $\{Y^{[MN]}\}$ that live in the adjoint representation of SO(8, $p$), with fundamental indices $M, N \in [\![1, 8 + p]\!]$. All the fields of the theory depend on the full higher-dimensional spacetime $\{x^\mu, Y^{[MN]}\}$, but the dependence is constrained by the section constraints

$$\begin{cases} \partial_{[MN} \otimes \partial_{PQ]} = 0\,, \\ \eta^{NP} \partial_{MN} \otimes \partial_{PQ} = 0\,, \end{cases} \tag{2.1}$$

with SO(8, $p$) invariant metric $\eta_{MN}$, which will be used in the following to raise and lower the internal indices. The notation $\otimes$ indicates that both derivative operators may act on different fields.

The bosonic fields of the theory are the following:

$$\left\{ g_{\mu\nu}, \mathcal{M}_{MN}, \mathcal{A}_\mu{}^{MN}, \mathcal{B}_{\mu MN} \right\}\,. \tag{2.2}$$

$g_{\mu\nu}$ describes the external metric, with signature $(-1, 1, 1)$, and $\mathcal{M}_{MN} \in$ SO(8, $p$) the internal metric. The vector fields $\mathcal{A}_\mu{}^{MN}$ and $\mathcal{B}_{\mu MN}$ are labeled by internal indices in the adjoint representation of SO(8, $p$). $\mathcal{B}_{\mu MN}$ is covariantly constrained: it has to satisfy algebraic constraints similar to Eq. (2.1) and compatibility conditions with the partial derivatives given by

$$\begin{cases} \mathcal{B}_{\mu[MN} \mathcal{B}_{\mu PQ]} = 0\,, \\ \eta^{NP} \mathcal{B}_{\mu MN} \mathcal{B}_{\mu PQ} = 0\,, \end{cases} \qquad \begin{cases} \mathcal{B}_{\mu[MN} \partial_{PQ]} = 0\,, \\ \eta^{NP} \mathcal{B}_{\mu MN} \partial_{PQ} = 0\,. \end{cases} \tag{2.3}$$

Its presence is necessary for the closure of the non-abelian gauge transformations [16].

## 2.1  Generalized internal diffeomorphisms and Lagrangian

The theory is invariant under local generalized internal diffeomorphisms, defined by their action on a vector $V^M$:

$$\mathcal{L}_{(\Lambda, \Sigma)} V^M = \Lambda^{KL} \partial_{KL} V^M + 2 \left( \partial^{KM} \Lambda_{KN} - \partial_{KN} \Lambda^{KM} + 2 \Sigma^M{}_N \right) V^N\,. \tag{2.4}$$

Their action on a tensor with an arbitrary number of fundamental SO(8, $p$) indices follows naturally. The gauge parameters $\Sigma_{MN}$ are subject to the same constraints as $\mathcal{B}_{\mu MN}$:

$$\begin{cases} \Sigma_{[MN} \Sigma_{PQ]} = 0\,, \\ \eta^{NP} \Sigma_{MN} \Sigma_{PQ} = 0\,, \end{cases} \qquad \begin{cases} \Sigma_{[MN} \partial_{PQ]} = 0\,, \\ \eta^{NP} \Sigma_{MN} \partial_{PQ} = 0\,. \end{cases} \tag{2.5}$$

The covariant external derivatives associated to such transformations are defined as

$$D_\mu = \partial_\mu - \mathcal{L}_{(\mathcal{A}_\mu, \mathcal{B}_\mu)}\,, \tag{2.6}$$

and ensure the invariance of the action.

---

[1]The theories considered in Ref. [16] are more general and duality-covariant with respect to O($p, q$). In addition to the series SO(8, $p$) of half-maximal theories, it includes in particular the series of theories based on SO(4, $p$), which reproduces the bosonic sector of certain quarter-maximal supergravities.

The full Lagrangian has the following form:

$$\mathscr{L} = \mathscr{L}_{\text{EH}} + \mathscr{L}_{\text{kin}} + \mathscr{L}_{\text{CS}} - \sqrt{-g}\, V \,. \tag{2.7}$$

$\mathscr{L}_{\text{EH}}$ is the modified Einstein-Hilbert term, defined in terms of the external dreibein $e_\mu{}^a$ and the covariantized Riemann tensor $\widehat{R}_{\mu\nu}{}^{ab}$:

$$\mathscr{L}_{\text{EH}} = \sqrt{-g}\, e_a{}^\mu e_b{}^\nu \left( \widehat{R}_{\mu\nu}{}^{ab} + F_{\mu\nu}{}^{MN} e^{a\rho}\, \partial_{MN} e_\rho{}^b \right), \tag{2.8}$$

where the Yang-Mills field strength $F_{\mu\nu}{}^{MN}$ has the expression

$$
\begin{aligned}
F_{\mu\nu}{}^{MN} = {} & 2\,\partial_{[\mu} \mathcal{A}_{\nu]}{}^{MN} - \mathcal{A}_{[\mu}{}^{KL} \partial_{KL} \mathcal{A}_{\nu]}{}^{MN} + \mathcal{A}_{[\mu}{}^{MN} \partial_{KL} \mathcal{A}_{\nu]}{}^{KL} \\
& + 4\, \mathcal{A}_{[\mu}{}^{K[M} \partial_{KL} \mathcal{A}_{\nu]}{}^{N]L} - 4\, \mathcal{A}_{[\mu}{}^{K[M} \partial^{N]L} \mathcal{A}_{\nu]KL} \,,
\end{aligned}
\tag{2.9}
$$

as implied by the commutator of Eq. (2.6). The scalar kinetic term has the usual form

$$\mathscr{L}_{\text{kin}} = \frac{1}{8} \sqrt{-g}\, D_\mu \mathcal{M}_{MN} D^\mu \mathcal{M}^{MN} \,, \tag{2.10}$$

and describes a $\text{SO}(8,p)/\big(\text{SO}(8) \times \text{SO}(p)\big)$ coset space $\sigma$-model. Finally, the Chern-Simons term is given by[2]

$$
\begin{aligned}
\mathscr{L}_{\text{CS}} = \sqrt{2}\, \varepsilon^{\mu\nu\rho} \Big( {} & F_{\mu\nu}{}^{MN} \mathcal{B}_{\rho\,MN} + \partial_\mu \mathcal{A}_{\nu N}{}^K \partial_{KM} \mathcal{A}_\rho{}^{MN} - \frac{2}{3} \partial_{MN} \partial_{KL} \mathcal{A}_\mu{}^{KP} \mathcal{A}_\nu{}^{MN} \mathcal{A}_{\rho\,P}{}^L \\
& + \frac{2}{3} A_\mu{}^{LN} \partial_{MN} \mathcal{A}_{\nu}{}^M{}_P \partial_{KL} \mathcal{A}_\rho{}^{PK} - \frac{4}{3} \mathcal{A}_\mu{}^{LN} \partial_{MP} \mathcal{A}_{\nu}{}^M{}_N \partial_{KL} \mathcal{A}_\rho{}^{PK} \Big),
\end{aligned}
\tag{2.11}
$$

and the so-called potential is bilinear in internal derivatives[3]:

$$
\begin{aligned}
V = {} & -\frac{1}{8} \partial_{KL} \mathcal{M}_{MN} \partial_{PQ} \mathcal{M}^{MN} \mathcal{M}^{KP} \mathcal{M}^{LQ} - \partial_{MK} \mathcal{M}^{NP} \partial_{NL} \mathcal{M}^{MQ} \mathcal{M}_{PQ} \mathcal{M}^{KL} \\
& + \frac{1}{4} \partial_{MN} \mathcal{M}^{PK} \partial_{KL} \mathcal{M}^{MQ} \mathcal{M}_P{}^L \mathcal{M}_Q{}^N + \partial_{MK} \mathcal{M}^{NK} \partial_{NL} \mathcal{M}^{ML} \\
& - g^{-1} \partial_{MN} g\, \partial_{KL} \mathcal{M}^{MK} \mathcal{M}^{NL} - \frac{1}{4} \mathcal{M}^{MK} \mathcal{M}^{NL} g^{-2} \partial_{MN} g\, \partial_{KL} g \\
& - \frac{1}{4} \mathcal{M}^{MK} \mathcal{M}^{NL} \partial_{MN} g_{\mu\nu} \partial_{KL} g^{\mu\nu} \,.
\end{aligned}
\tag{2.12}
$$

Once restricted to a solution of the section constraints (2.1), this theory (2.7) describes higher-dimensional supergravity.

## 2.2 Generalized Scherk-Schwarz ansatz

One of the main achievements of exceptional field theories is the construction of consistent truncations [20–22]. These truncations can be defined by a generalized Scherk-Schwarz compactification ansatz that encodes the dependence of the fields on the internal coordinates in a twist matrix and a weight factor $\rho(Y)$. The twist matrix is an $\text{SO}(8,p)$-valued matrix $U_M{}^{\bar N}(Y)$, so that the compactification ansätze for the fields take the form [16]

$$
\begin{cases}
g_{\mu\nu}(x,Y) & = \rho(Y)^{-2} g_{\mu\nu}(x), \\
\mathcal{M}_{MN}(x,Y) & = U_M{}^{\bar M}(Y) U_N{}^{\bar N}(Y) \mathcal{M}_{\bar M \bar N}(x), \\
\mathcal{A}_\mu{}^{MN}(x,Y) & = \rho(Y)^{-1} U^M{}_{\bar M}(Y) U^N{}_{\bar N}(Y) \mathcal{A}_\mu{}^{\bar M \bar N}(x), \\
\mathcal{B}_{\mu MN}(x,Y) & = -\frac{1}{4} \rho(Y)^{-1} U^K{}_{\bar N}(Y) \partial_{MN} U_{K\bar M}(Y) \mathcal{A}_\mu{}^{\bar M \bar N}(x).
\end{cases}
\tag{2.13}
$$

---

[2] $\varepsilon^{\mu\nu\rho}$ denotes the constant Levi-Civita density.

[3] This expression differs from the one given in Ref. [16]: we have corrected some coefficients.

The indices $\bar{M} \in [\![1, 8+p]\!]$ are flat, fundamental $SO(8, p)$ indices and describe three-dimensional quantities. All the curved information of the internal manifold is encoded in the twist matrix, and the fields that depend on the external coordinates only are the fields of three-dimensional gauged supergravity. We also consider gauged parameters of the following form:

$$
\begin{cases}
\Lambda^{MN}(x, Y) &= \rho(Y)^{-1} U^M{}_{\bar{M}}(Y) U^N{}_{\bar{N}}(Y) \Lambda^{\bar{M}\bar{N}}(x), \\
\Sigma_{MN}(x, Y) &= -\dfrac{1}{4} \rho(Y)^{-1} U^K{}_{\bar{N}}(Y) \partial_{MN} U_{K\bar{M}}(Y) \Lambda^{\bar{M}\bar{N}}(x).
\end{cases}
\tag{2.14}
$$

The consistency of the truncation can then be written in terms of differential equations on the weight factor and the twist matrix. Defining the embedding tensor

$$
X_{\bar{M}\bar{N}|\bar{P}\bar{Q}} = \theta_{\bar{M}\bar{N}\bar{P}\bar{Q}} + \frac{1}{2}\left(\eta_{\bar{P}[\bar{M}}\theta_{\bar{N}]\bar{Q}} - \eta_{\bar{Q}[\bar{M}}\theta_{\bar{N}]\bar{P}}\right) + \theta\,\eta_{\bar{P}[\bar{M}}\eta_{\bar{N}]\bar{Q}},
\tag{2.15}
$$

with components

$$
\begin{aligned}
\theta_{\bar{M}\bar{N}\bar{P}\bar{Q}} &= 6\,\rho^{-1}\,\partial_{PQ} U_{M[\bar{M}} U^M{}_{\bar{N}} U^P{}_{\bar{P}} U^Q{}_{\bar{Q}]}, \\
\theta_{\bar{M}\bar{N}} &= 4\rho^{-1}\,U^M{}_{\bar{M}}\partial_{MN}U^N{}_{\bar{N}} - \eta_{\bar{M}\bar{N}}\,\theta - 4\rho^{-2}\,U^M{}_{\bar{M}}U^N{}_{\bar{N}}\partial_{MN}\rho, \\
\theta &= \frac{4\rho^{-1}}{8+p}\,U^{P\bar{Q}}\partial_{PQ}U^Q{}_{\bar{Q}},
\end{aligned}
\tag{2.16}
$$

the consistency of the truncation is ensured if all the components of the embedding tensor are constant: all dependences on the internal coordinates in the equations of motion are then factored out, and $X_{\bar{M}\bar{N}|\bar{P}\bar{Q}}$ captures the gauge structure of the three-dimensional theory. The quadratic constraint

$$
X_{\bar{K}\bar{L}|\bar{P}}{}^{\bar{R}} X_{\bar{M}\bar{N}|\bar{R}}{}^{\bar{Q}} - X_{\bar{M}\bar{N}|\bar{P}}{}^{\bar{R}} X_{\bar{K}\bar{L}|\bar{R}}{}^{\bar{Q}} = 2 X_{\bar{K}\bar{L}|[\bar{M}}{}^{\bar{R}} X_{\bar{N}]\bar{R}|\bar{P}}{}^{\bar{Q}},
\tag{2.17}
$$

needed to ensure the closure of the gauged algebra [14], is automatically satisfied thanks to the section constraints (2.1). The consistency conditions for the twist matrix and the weight factor can also be expressed as

$$
\mathcal{L}_{(\Lambda^{MN}, \Sigma_{MN})} U^P{}_{\bar{P}} = \Lambda^{\bar{M}\bar{N}} X_{\bar{M}\bar{N}|\bar{P}}{}^{\bar{Q}} U^P{}_{\bar{Q}},
\tag{2.18}
$$

and accordingly as conditions of generalized parallelizability [16].

We further impose the tensor $\theta_{\bar{M}\bar{N}}$ defined in Eq. (2.16) to be symmetric. Indeed, if its anti-symmetric part is non-vanishing, the three-dimensional field equations include a gauging of the trombone scaling symmetry [16] and, in turn, the resulting theory does not admit a three-dimensional action. Let us finally note that the definition (2.16) together with the constraints (2.1) imposes

$$
\theta_{[\bar{K}\bar{L}\bar{M}\bar{N}}\theta_{\bar{P}\bar{Q}\bar{R}\bar{S}]} = 0.
\tag{2.19}
$$

Thus, the only gaugings that can be reproduced by this generalized Scherk-Schwarz procedure are those which satisfy this additional constraint. This is consistent with the fact that the potential (2.12) cannot produce terms proportional to $\theta_{[\bar{K}\bar{L}\bar{M}\bar{N}}\theta_{\bar{P}\bar{Q}\bar{R}\bar{S}]}$, whereas the most general potential of three-dimensional half-maximal gauged supergravity carries such a term [23].

## 3 Kaluza-Klein spectroscopy

We consider a fixed $AdS_3 \times \mathcal{M}$ supergravity background, with internal manifold $\mathcal{M}$, which in the three-dimensional supergravity variables takes the diagonal form

$$
\{g_{\mu\nu} = \mathring{g}_{\mu\nu}, \mathcal{M}_{\bar{M}\bar{N}} = \Delta_{\bar{M}\bar{N}}, \mathcal{A}_\mu{}^{\bar{M}\bar{N}} = 0\}.
\tag{3.1}
$$

To compute the full Kaluza-Klein spectrum around this background, we need to consider linear fluctuations which we expand in terms of a basis of the fields on the internal manifold. To do so, we take profit of the powerful ansätze of Ref. [12]: by introducing the fluctuations directly in the exceptional field theory ansätze (2.13), all the tensorial structure of the fields is factored out, so that they are scalars on the internal manifold. We then only need a basis of scalar harmonics $\mathcal{Y}^{\Sigma}$. We thus consider the following linear fluctuations:

$$\begin{cases} g_{\mu\nu}(x,Y) & = \rho(Y)^{-2}\big(\mathring{g}_{\mu\nu}(x) + \mathcal{Y}^{\Sigma}(Y)\,\mathring{g}_{\mu\nu}{}^{\Sigma}(x)\big)\,, \\[2mm] \mathcal{M}_{MN}(x,Y) & = U_M{}^{\bar{M}}(Y)U_N{}^{\bar{N}}(Y)\big(\Delta_{\bar{M}\bar{N}} + \mathcal{Y}^{\Sigma}(Y)\,j_{\bar{M}\bar{N}}{}^{\Sigma}(x)\big)\,, \\[2mm] \mathcal{A}_{\mu}{}^{MN}(x,Y) & = \rho(Y)^{-1}U^M{}_{\bar{M}}(Y)U^N{}_{\bar{N}}(Y)\mathcal{Y}^{\Sigma}(Y)A_{\mu}{}^{\bar{M}\bar{N},\Sigma}(x)\,, \\[2mm] \mathcal{B}_{\mu\,MN}(x,Y) & = -\dfrac{1}{4}\rho(Y)^{-1}U^K{}_{\bar{N}}(Y)\partial_{MN}U_{K\bar{M}}(Y)\mathcal{Y}^{\Sigma}(Y)A_{\mu}{}^{\bar{M}\bar{N},\Sigma}(x)\,. \end{cases} \tag{3.2}$$

For the internal metric $\mathcal{M}_{MN}$ to belong to SO(8, $p$), the scalar fluctuations $j_{\bar{M}\bar{N}}{}^{\Sigma}$ are such that

$$\Delta_{\bar{P}(\bar{M}}\,\eta^{\bar{P}\bar{Q}}j_{\bar{N})\bar{Q}}{}^{\Sigma} = 0\,. \tag{3.3}$$

Note that the fluctuation for $\mathcal{B}_{\mu\,MN}$ are not independent form the ones of $\mathcal{A}_{\mu}{}^{MN}$. This is motivated by the generalized Scherk-Schwarz ansatz (2.13), where the ansätze for these fields are based on the same three-dimensional field $\mathcal{A}_{\mu}{}^{\bar{M}\bar{N}}$. We will see in the following that the consistency of the linearized equations of motion precisely requires this structure of the fluctuations.

As the topology of the compactification is the same for any solution of the three-dimensional theory, we consider harmonics that form representations of the largest symmetry group possible, noted $G_{max}$ (with transitive action on the coset space), which corresponds to the maximally supersymmetric point of the three-dimensional gauged supergravity [12]. We restrict ourselves to theories with compact $G_{max}$. The action of the internal derivatives on the scalar harmonics is then given by

$$\rho^{-1}U^M{}_{\bar{M}}\,U^N{}_{\bar{N}}\,\partial_{MN}\mathcal{Y}^{\Sigma} = -\mathcal{T}_{\bar{M}\bar{N}}{}^{\Sigma\Omega}\mathcal{Y}^{\Omega}\,. \tag{3.4}$$

The matrices $\mathcal{T}_{\bar{M}\bar{N}}{}^{\Sigma\Omega} = -\mathcal{T}_{\bar{M}\bar{N}}{}^{\Omega\Sigma}$ correspond to the generators of $G_{max}$ in the representation of the scalar harmonics. They are normalized with respect to the embedding tensor:

$$\big[\mathcal{T}_{\bar{M}\bar{N}}, \mathcal{T}_{\bar{P}\bar{Q}}\big] = -X_{\bar{M}\bar{N}|[\bar{P}}{}^{\bar{K}}\mathcal{T}_{\bar{Q}]\bar{K}} + X_{\bar{P}\bar{Q}|[\bar{M}}{}^{\bar{K}}\mathcal{T}_{\bar{N}]\bar{K}}\,. \tag{3.5}$$

We use in the following the ansätze (3.2) to compute the mass matrices around the background (3.1). The Kaluza-Klein towers will contain massive spin-2 fields and massive vectors, which are, respectively, induced by Goldstone modes in the vectors and scalars spectra *via* Brout-Englert-Higgs (BEH) mechanisms: each massive spin-2 field absorbs a vector and a scalar, all representatives of the same representation, and each massive vector absorbs a massless scalar, also in the same representation. These modes have to be eliminated from the spectra calculated from the mass matrices given below. We refer to Ref. [12] for a complete account of these effects.

## 3.1 Spin-2 fields

The mass matrix for the spin-2 fields can be computed in the standard supergravity formulation by solving a wave equation on the internal space [6]. In the context of exceptional field theory, it features a universal form [12, 24], which we simply reproduce here:

$$M_{(2)}^2{}^{\Sigma\Omega} = -\Delta^{\bar{M}\bar{P}}\Delta^{\bar{N}\bar{Q}}\mathcal{T}_{\bar{M}\bar{N}}{}^{\Sigma\Gamma}\mathcal{T}_{\bar{P}\bar{Q}}{}^{\Gamma\Omega}\,. \tag{3.6}$$

In three dimensions each eigenstate of $M_{(2)}^2{}^{\Sigma\Omega}$ gives rise to two degrees of freedom, one with helicity $s = 2$ and one with helicity $s = -2$.

## 3.2 Vector mass matrix

To compute the vector mass matrix, we start from the variation of the Lagrangian (2.7) with respect to the vectors $\mathcal{A}_\mu{}^{MN}$ and $\mathcal{B}_{\mu MN}$

$$\delta_{(\mathcal{A},\mathcal{B})}\mathscr{L} = \varepsilon^{\mu\nu\rho}\left(\mathcal{E}^{(\mathcal{A})MN}_{\mu\nu}\,\delta\mathcal{B}_{\rho\,MN} + \mathcal{E}^{(\mathcal{B})}_{\mu\nu MN}\,\delta\mathcal{A}_\rho{}^{MN}\right), \tag{3.7}$$

where [16]

$$\begin{aligned}
\mathcal{E}^{(\mathcal{A})MN}_{\mu\nu} &= \sqrt{2}\,F_{\mu\nu}{}^{MN} - \sqrt{-g}\,\varepsilon_{\mu\nu\rho}\,\mathrm{j}^{\rho\,MN}\,,\\
\mathcal{E}^{(\mathcal{B})}_{\mu\nu MN} &= \sqrt{2}\,G_{\mu\nu MN} + \sqrt{-g}\,\varepsilon_{\mu\nu\rho}\,J^\rho{}_{MN} - \frac{1}{8}\sqrt{-g}\,\varepsilon_{\mu\nu\rho}\,\mathrm{j}^{\rho\,K}{}_L\,\mathcal{J}_{MN}{}^L{}_K + \partial_{MK}\mathcal{E}^{(\mathcal{A})}_{\mu\nu}{}_N{}^K\,.
\end{aligned} \tag{3.8}$$

The field strength $F_{\mu\nu}{}^{MN}$ has been given in Eq. (2.9). $G_{\mu\nu MN}$ and the different currents are defined as follows:

$$G_{\mu\nu MN} = 2\,D_{[\mu}\mathcal{B}_{\nu]MN} - \mathcal{A}_{[\mu K}{}^P\partial_{PQ}\partial_{MN}\mathcal{A}_{\nu]}{}^{KQ}\,, \tag{3.9}$$

$$\mathcal{J}_{MN,KL} = \partial_{MN}\mathcal{M}_{LP}\,\mathcal{M}^P{}_K\,, \tag{3.10}$$

$$\mathrm{j}_\mu{}^{MN} = \eta_{KL}\mathcal{M}^{K[M}D_\mu\mathcal{M}^{N]L}\,, \tag{3.11}$$

$$J^\mu{}_{MN} = -2\,e^\mu{}_a e^\nu{}_b\left[\partial_{MN}\omega_\nu{}^{ab} - D_\nu\left(e^{\rho[a}\partial_{MN}e_\rho{}^{b]}\right)\right], \tag{3.12}$$

with the spin connection $\omega_\nu{}^{ab}$. Injecting the fluctuations (3.2) in Eq. (2.9) and (3.9)–(3.12) and considering the linearization with respect to the fluctuation $A_\mu{}^{\bar{M}\bar{N},\Sigma}$, we get

$$F_{\mu\nu}{}^{MN} \underset{\text{lin.}}{=} \rho^{-1}U^M{}_{\bar{M}}U^N{}_{\bar{N}}\,2\,\partial_{[\mu}A_{\nu]}{}^{\bar{M}\bar{N},\Sigma}\,\mathcal{Y}^\Sigma\,, \tag{3.13}$$

$$G_{\mu\nu MN} \underset{\text{lin.}}{=} -\frac{1}{2}\rho^{-1}U^K{}_{\bar{N}}\partial_{MN}U_{K\bar{M}}\,\partial_{[\mu}A_{\nu]}{}^{\bar{M}\bar{N},\Sigma}\,\mathcal{Y}^\Sigma\,, \tag{3.14}$$

$$\mathrm{j}_\mu{}^{MN} \underset{\text{lin.}}{=} U^{[M}{}_{\bar{M}}U^{N]}{}_{\bar{N}}\left(\Delta^{\bar{M}\bar{K}}\Delta^{\bar{N}\bar{L}} - \eta^{\bar{M}\bar{K}}\eta^{\bar{N}\bar{L}}\right)\left[X_{\bar{K}\bar{L}|\bar{P}\bar{Q}}\delta^{\Sigma\Omega} + 4\mathcal{T}_{\bar{P}\bar{K}}{}^{\Sigma\Omega}\eta_{\bar{L}\bar{Q}}\right]A_\mu{}^{\bar{P}\bar{Q},\Sigma}\,\mathcal{Y}^\Omega\,. \tag{3.15}$$

Thus, once linearized, the variation (3.7) takes the form

$$\begin{aligned}
\delta_{(\mathcal{A},\mathcal{B})}\mathscr{L} \underset{\text{lin.}}{=} &-\frac{1}{2\sqrt{2}}\varepsilon^{\mu\nu\rho}\,\rho^{-1}\left(X_{\bar{M}\bar{N}|\bar{U}\bar{V}}\delta^{\Sigma\Omega} + 4\mathcal{T}_{\bar{U}[\bar{M}}{}^{\Sigma\Omega}\eta_{\bar{N}]\bar{V}}\right)\delta A_\rho{}^{\bar{U}\bar{V},\Delta}\,\mathcal{Y}^\Delta\mathcal{Y}^\Omega\\
&\times\left[2\,\partial_{[\mu}A_{\nu]}{}^{\bar{M}\bar{N},\Sigma} + \sqrt{-\mathring{g}}\,\varepsilon_{\mu\nu\sigma}\,M_{(1)}{}^{\bar{M}\bar{N}\,\Sigma}{}_{\bar{P}\bar{Q}}{}^\Gamma A^{\sigma\,\bar{P}\bar{Q},\Gamma}\right],
\end{aligned} \tag{3.16}$$

with the mass matrix of the vector fields

$$M_{(1)}{}^{\bar{M}\bar{N}\,\Sigma}{}_{\bar{P}\bar{Q}}{}^\Omega = \frac{1}{\sqrt{2}}\left(\eta^{\bar{K}[\bar{M}}\eta^{\bar{N}]\bar{L}} - \Delta^{\bar{K}[\bar{M}}\Delta^{\bar{N}]\bar{L}}\right)\left(X_{\bar{K}\bar{L}|\bar{P}\bar{Q}}\delta^{\Sigma\Omega} + 4\mathcal{T}_{\bar{K}[\bar{P}}{}^{\Sigma\Omega}\eta_{\bar{Q}]\bar{L}}\right). \tag{3.17}$$

As the equations of motion are of first order, each eigenstate of the mass matrix gives rise to a single degree of freedom, whose helicity is given by the sign of its eigenvalue.

The second line of Eq. (3.16) is the equation of motion of a topologically massive vector in three dimensions. In absence of the $\mathcal{T}$ tensors, it reproduces the Scherk-Schwarz reduction to three dimensions. The $\mathcal{T}$ tensors capture the effect of internal derivatives on the harmonics. In Eq. (3.16), the equation of motion is further contracted with another mass matrix (3.17). This imposes that the eigenvectors of $M_{(1)}{}^{\bar{M}\bar{N}\,\Sigma}{}_{\bar{P}\bar{Q}}{}^\Omega$ with vanishing eigenvalues are projected out of the equation of motion, and do not belong to the physical spectrum.

### 3.3 Scalar mass matrix

The computation of the scalar mass matrix, though more involved, follows the same steps as the ones of the vector mass matrix. First, the variation of the Lagrangian (2.7) with respect to $\mathcal{M}_{MN}$ has the form

$$\delta_{\mathcal{M}}\mathscr{L} = \mathcal{K}_{MN}^{(\mathcal{M})}\,\delta\mathcal{M}^{MN}\,. \tag{3.18}$$

As $\mathcal{M}_{MN} \in \mathrm{SO}(8,p)$, it is a constrained field and one has to project $\mathcal{K}_{MN}^{(\mathcal{M})}$ onto symmetric coset valued indices to produce the equations of motion.

It remains then to inject the fluctuations (3.2) in $\mathcal{K}_{MN}^{(\mathcal{M})}$ and to linearize with respect to $j_{\bar{M}\bar{N}}{}^{\Sigma}$. Contrary to the vectors, the equations of motion of the scalars are however of second order in internal derivatives, which complicates considerably the task of factoring out the dependence on the internal coordinates. The computation is however made easier by adopting the following strategy [12]: when an internal derivative hits the linear fluctuations (3.2), it produces derivatives $\partial U$ and $\partial\rho$ of the twist matrix and the weight factor, which will contribute to the embedding tensor (2.16), as well as derivatives $\partial\mathcal{Y}$ of the harmonics, which will form $\mathcal{T}$ tensors following Eq. (3.4). As the equation of motion is of second order in internal derivatives, the squared scalar mass matrix will be schematically organized into

$$M_{(0)}^2 = \theta\theta + \theta\mathcal{T} + \mathcal{T}\mathcal{T}\,. \tag{3.19}$$

The $\theta\theta$ term is given by construction by the scalar mass matrix of the three dimensional gauged supergravity, and it can be extracted from the three dimensional potential [23][4]

$$
\begin{aligned}
V_{3d} = &\frac{1}{24}\,\theta_{\bar{K}\bar{L}\bar{M}\bar{N}}\,\theta_{\bar{P}\bar{Q}\bar{R}\bar{S}}\Big(\Delta^{\bar{K}\bar{P}}\Delta^{\bar{L}\bar{Q}}\Delta^{\bar{M}\bar{R}}\Delta^{\bar{N}\bar{Q}} - 6\,\Delta^{\bar{K}\bar{P}}\Delta^{\bar{L}\bar{Q}}\eta^{\bar{M}\bar{R}}\eta^{\bar{N}\bar{Q}} \\
&\qquad\qquad\qquad + 8\,\Delta^{\bar{K}\bar{P}}\eta^{\bar{L}\bar{Q}}\eta^{\bar{M}\bar{R}}\eta^{\bar{N}\bar{Q}} - 3\,\eta^{\bar{K}\bar{P}}\eta^{\bar{L}\bar{Q}}\eta^{\bar{M}\bar{R}}\eta^{\bar{N}\bar{Q}}\Big) \\
&+ \frac{1}{16}\,\theta_{\bar{K}\bar{L}}\,\theta_{\bar{P}\bar{Q}}\Big(2\,\Delta^{\bar{K}\bar{P}}\Delta^{\bar{L}\bar{Q}} - 2\,\eta^{\bar{K}\bar{P}}\eta^{\bar{L}\bar{Q}} - \Delta^{\bar{K}\bar{L}}\Delta^{\bar{P}\bar{Q}}\Big) + 2\,\theta\,\theta_{\bar{K}\bar{L}}\,\Delta^{\bar{K}\bar{L}} - 16\,\theta^2\,.
\end{aligned} \tag{3.20}
$$

We could then focus on the remaining terms while linearizing the equation of motion and injecting the fluctuations ansätze. As we are not considering the $\theta\theta$ term, it is sufficient to contract the linearization of $\mathcal{K}_{MN}^{(\mathcal{M})}$ with $j^{MN,\Sigma} = U^{M\bar{M}}U^{N\bar{N}}j_{\bar{M}\bar{N}}{}^{\Sigma}$ to restrict ourselves on symmetric coset valued indices. For example, the first term of the potential (2.12) contributes to $\mathcal{K}_{MN}^{(\mathcal{M})}$ with a term

$$\frac{1}{4}\,\sqrt{-g}\,\partial_{ML}\mathcal{M}_{KP}\,\partial_{NQ}\mathcal{M}^{KP}\mathcal{M}^{LQ}\,, \tag{3.21}$$

which, once linearized and projected onto symmetric coset valued indices, gives

$$
\begin{aligned}
\frac{1}{4}\,\sqrt{-g}\,j^{MN,\Sigma}\,&\partial_{ML}\mathcal{M}_{KP}\,\partial_{NQ}\mathcal{M}^{KP}\mathcal{M}^{LQ} \\
&\underset{\text{lin.}}{=} -\sqrt{-\mathring{g}}\,\rho^{-1}j^{\bar{M}\bar{N},\Sigma}j^{\bar{P}\bar{Q},\Omega}\mathcal{Y}^{\Delta}\,J_{\bar{R}\bar{P}|\bar{M}\bar{K}}\,\Delta_{\bar{Q}}{}^{\bar{R}}\Delta^{\bar{K}\bar{L}}\mathcal{T}_{\bar{N}\bar{L}}{}^{\Omega\Delta} + (\dots)\,,
\end{aligned} \tag{3.22}
$$

where we noted $J_{\bar{K}\bar{L}|\bar{P}\bar{Q}} = \rho^{-1}\partial_{PQ}U_{K\bar{K}}\,U^{K}{}_{\bar{L}}U^{P}{}_{\bar{P}}U^{Q}{}_{\bar{Q}}$. The ellipses denote the terms which do not contribute to the $\theta\mathcal{T} + \mathcal{T}\mathcal{T}$ terms. After considering all the terms in $\mathcal{K}_{MN}^{(\mathcal{M})}$ and restoring the $\theta\theta$ terms, the linearization finally results in the following mass matrix:

$$M_{(0)\bar{M}\bar{N}}{}^{\Sigma}{}_{\bar{P}\bar{Q}}{}^{\Omega}\,j^{\bar{M}\bar{N},\Sigma}j^{\bar{P}\bar{Q},\Omega} = \big(m_{\bar{M}\bar{N},\bar{P}\bar{Q}}\,\delta^{\Sigma\Omega} + m'_{\bar{M}\bar{N}}{}^{\Sigma}{}_{\bar{P}\bar{Q}}{}^{\Omega}\big)\,j^{\bar{M}\bar{N},\Sigma}j^{\bar{P}\bar{Q},\Omega}\,, \tag{3.23}$$

---

[4]Following Ref. [25], a typo has been corrected in the second line. We have also omitted a term proportional to $\theta_{[\bar{K}\bar{L}\bar{M}\bar{N}}\theta_{\bar{P}\bar{Q}\bar{R}\bar{S}]}$, as all the embedding tensors we are interested in are obtained by generalized Scherk-Schwarz reduction and as such satisfy Eq. (2.19).

where

$$
\begin{aligned}
m_{\bar{M}\bar{N},\bar{P}\bar{Q}} = {} & 2\,\theta_{\bar{M}\bar{P}\bar{K}\bar{L}}\,\theta_{\bar{N}\bar{Q}\bar{R}\bar{S}}\,\Delta^{\bar{K}\bar{R}}\Delta^{\bar{L}\bar{S}} + \frac{2}{3}\,\theta_{\bar{M}\bar{U}\bar{K}\bar{L}}\,\theta_{\bar{P}\bar{V}\bar{R}\bar{S}}\,\delta_{\bar{N}\bar{Q}}\,\Delta^{\bar{U}\bar{V}}\Delta^{\bar{K}\bar{R}}\Delta^{\bar{L}\bar{S}} \\
& - 2\,\theta_{\bar{M}\bar{P}\bar{K}\bar{L}}\,\theta_{\bar{N}\bar{Q}}{}^{\bar{K}\bar{L}} - 2\,\theta_{\bar{M}\bar{U}\bar{K}\bar{L}}\,\theta_{\bar{P}\bar{V}}{}^{\bar{K}\bar{L}}\delta_{\bar{N}\bar{Q}}\,\Delta^{\bar{U}\bar{V}} + \frac{4}{3}\,\theta_{\bar{M}\bar{U}\bar{K}\bar{L}}\,\theta_{\bar{P}}{}^{\bar{U}\bar{K}\bar{L}}\delta_{\bar{N}\bar{Q}} \\
& + \theta_{\bar{M}\bar{P}}\,\theta_{\bar{N}\bar{Q}} - \frac{1}{2}\,\theta_{\bar{M}\bar{N}}\,\theta_{\bar{P}\bar{Q}} + \theta_{\bar{M}\bar{K}}\,\theta_{\bar{P}\bar{L}}\,\delta_{\bar{N}\bar{Q}}\,\Delta^{\bar{K}\bar{L}} \\
& - \frac{1}{2}\,\theta_{\bar{M}\bar{P}}\,\theta_{\bar{K}\bar{L}}\,\delta_{\bar{N}\bar{Q}}\,\Delta^{\bar{K}\bar{L}} + 8\,\theta\,\theta_{\bar{M}\bar{P}}\,\delta_{\bar{N}\bar{Q}}\,,
\end{aligned}
\tag{3.24}
$$

$$
\begin{aligned}
m'_{\bar{M}\bar{N}}{}^{\Sigma}{}_{\bar{P}\bar{Q}}{}^{\Omega} = {} & 4\,\theta_{\bar{M}\bar{P}\bar{R}\bar{K}}\,\Delta_{\bar{N}}{}^{\bar{R}}\Delta^{\bar{K}\bar{L}}\,\mathcal{T}_{\bar{Q}\bar{L}}{}^{\Sigma\Omega} + 4\,\theta_{\bar{M}\bar{P}\bar{R}\bar{K}}\,\Delta_{\bar{Q}}{}^{\bar{R}}\Delta^{\bar{K}\bar{L}}\,\mathcal{T}_{\bar{N}\bar{L}}{}^{\Sigma\Omega} \\
& - 4\,\eta_{\bar{M}\bar{P}}\,\theta_{\bar{N}\bar{Q}\bar{K}\bar{L}}\,\Delta^{\bar{K}\bar{R}}\Delta^{\bar{L}\bar{S}}\,\mathcal{T}_{\bar{R}\bar{S}}{}^{\Sigma\Omega} + 4\,\eta_{\bar{M}\bar{P}}\,\theta_{\bar{N}\bar{Q}\bar{K}\bar{L}}\,\mathcal{T}^{\bar{K}\bar{L}\,\Sigma\Omega} \\
& + 4\left(\theta_{\bar{M}\bar{P}} + \theta\,\eta_{\bar{M}\bar{P}}\right)\mathcal{T}_{\bar{N}\bar{Q}}{}^{\Sigma\Omega} + \eta_{\bar{M}\bar{P}}\,\eta_{\bar{N}\bar{Q}}\,\Delta^{\bar{K}\bar{R}}\Delta^{\bar{L}\bar{S}}\,\mathcal{T}_{\bar{K}\bar{L}}{}^{\Sigma\Lambda}\mathcal{T}_{\bar{R}\bar{S}}{}^{\Lambda\Omega} \\
& + 8\,\Delta_{\bar{M}\bar{P}}\,\Delta^{\bar{K}\bar{L}}\,\mathcal{T}_{\bar{Q}\bar{L}}{}^{\Sigma\Lambda}\mathcal{T}_{\bar{N}\bar{K}}{}^{\Lambda\Omega} - 2\,\Delta_{\bar{M}}{}^{\bar{K}}\Delta_{\bar{P}}{}^{\bar{L}}\,\mathcal{T}_{\bar{Q}\bar{L}}{}^{\Sigma\Lambda}\mathcal{T}_{\bar{N}\bar{K}}{}^{\Lambda\Omega} + 8\,\mathcal{T}_{\bar{M}\bar{P}}{}^{\Sigma\Lambda}\mathcal{T}_{\bar{N}\bar{Q}}{}^{\Lambda\Omega}\,.
\end{aligned}
\tag{3.25}
$$

### 3.4 Spectra and conformal dimensions

The masses of the bosons in the Kaluza-Klein spectrum are given by the eigenvalues $m_{(2)}^2$, $m_{(1)}$ and $m_{(0)}^2$ of the matrices (3.6), (3.17) and (3.23). In the context of holography, the spectrum is then most conveniently given in terms of the corresponding conformal dimensions $\Delta_{(s)}$. If the vacuum preserves some supersymmetries, they allow the identification of the supermultiplets. In three dimensions, the conformal dimensions are related to the normalized masses through [26,27]

$$
\Delta_{(2)}\left(\Delta_{(2)} - 2\right) = \left(m_{(2)}\ell_{\mathrm{AdS}}\right)^2, \quad \Delta_{(1)} = 1 + |m_{(1)}\ell_{\mathrm{AdS}}|, \quad \Delta_{(0)}\left(\Delta_{(0)} - 2\right) = \left(m_{(0)}\ell_{\mathrm{AdS}}\right)^2,
\tag{3.26}
$$

where the masses are normalized by the AdS length $\ell_{\mathrm{AdS}} = \sqrt{2/|V_0|}$ with $V_0$ the potential (3.20) at the vacuum. Upon projecting out the Goldstone vectors and scalars, the spectrum organizes into multiplets of the (super)group $\mathcal{G}$ that describes the isometries of the (super)symmetric AdS$_3$ vacuum. The isometry group of AdS$_3$ is $\mathrm{SO}(2,2) \cong \mathrm{SL}(2,\mathbb{R}) \times \mathrm{SL}(2,\mathbb{R})$ and is not simple, so that $\mathcal{G}$ is a direct product $\mathcal{G}_{\mathrm{L}} \times \mathcal{G}_{\mathrm{R}}$ of simple (super)groups. If the vacuum is supersymmetric, the even parts of $\mathcal{G}_{\mathrm{L}}$ and $\mathcal{G}_{\mathrm{R}}$ are isomorphic to the product of an $R$-symmetry group and an AdS$_3$ factor $\mathrm{SL}(2,\mathbb{R})$. Such supergroups have been classified in Ref. [28] and further studied Ref. [29]. Supersymmetry in three dimensions is thus factorizable and decomposes into $\mathcal{N} = (\mathcal{N}_{\mathrm{L}}, \mathcal{N}_{\mathrm{R}})$, where $\mathcal{N}_{\mathrm{L,R}}$ denote the number of fermionic generators of $\mathcal{G}_{\mathrm{L,R}}$. The conformal dimension $\Delta$ also decomposes itself as $\Delta = \Delta_{\mathrm{L}} + \Delta_{\mathrm{R}}$, with conformal dimensions $\Delta_{\mathrm{L,R}}$ associated to the representations of $\mathcal{G}_{\mathrm{L,R}}$. The spacetime spin $s$ is identified as $s = \Delta_{\mathrm{R}} - \Delta_{\mathrm{L}}$, and the couples $(\Delta, s)$ then label the representations of the AdS$_3$ group $\mathrm{SO}(2,2)$.

## 4 Examples

We illustrate in the following the tools developed in the previous section using three examples: $\mathcal{N}_{\mathrm{6d}} = (2,0)$ and $\mathcal{N}_{\mathrm{6d}} = (1,1)$ six-dimensional supergravities[5] on AdS$_3 \times S^3$, and ten-dimensional supergravity on AdS$_3 \times S^3 \times S^3 \times S^1$.

---

[5]$\mathcal{N}_{\mathrm{6d}}$ denotes the number of supersymmetries for theories in six dimensions, and should not be mixed up with its three-dimensional analogue $\mathcal{N}$.

Tab. 1: Short multiplet $\boldsymbol{k+1}$ of SU$(2|1,1)_{\text{L,R}}$ for $k \geq 2$ [17]. The multiplet $\boldsymbol{2}$ is obtained by suppressing the first line for $k = 1$, and $\boldsymbol{1}$ the two first lines for $k = 0$.

| $\Delta_{\text{L,R}}$ | SU$(2)_{\text{L,R global}} \times$ SU$(2)_{\text{L,R gauge}}$ |
|:---:|:---:|
| $(k+2)/2$ | $\big(0, (k-2)/2\big)$ |
| $(k+1)/2$ | $\big(1/2, (k-1)/2\big)$ |
| $k/2$ | $\big(0, k/2\big)$ |

## 4.1 Six-dimensional supergravities on AdS$_3 \times S^3$

Six-dimensional $\mathcal{N}_{6d} = (1,0)$ supergravity coupled to a tensor multiplet admits a consistent truncation on the sphere $S^3$ [30, 31]. The reduction gives rise to a three-dimensional theory, whose scalars parametrize the coset space SO$(4,4)/\big(\text{SO}(4) \times \text{SO}(4)\big)$, so that the truncation can be described in terms of SO$(4,4)$ exceptional field theory [16]. The theory in six dimensions features an AdS$_3 \times S^3$ vacuum that preserves $\mathcal{N}_{6d} = (1,0)$ supersymmetry.

This six-dimensional theory can be embedded into half-maximal $\mathcal{N}_{6d} = (2,0)$ and $\mathcal{N}_{6d} = (1,1)$ supergravities[6]. The AdS$_3 \times S^3$ vacuum then preserves all the supersymmetries in the case $\mathcal{N}_{6d} = (2,0)$, but only half of them within $\mathcal{N}_{6d} = (1,1)$. The associated three-dimensional theories have an SO$(4)$ gauge group and scalars organized in an SO$(8,4)/\big(\text{SO}(8) \times \text{SO}(4)\big)$ coset space, and their potentials possess stable supersymmetric AdS$_3$ vacua preserving $\mathcal{N} = (4,4)$ and $\mathcal{N} = (0,4)$ supersymmetries, respectively. Couplings to $m$ other tensor multiplets can be added in six-dimensions for $\mathcal{N}_{6d} = (2,0)$, and similarly to $m$ vector multiplets for $\mathcal{N}_{6d} = (1,1)$, leading in the exceptional field theory description to a coset space SO$(8,4+m)/\big(\text{SO}(8) \times \text{SO}(4+m)\big)$. The descriptions of the associated consistent truncations in terms of generalized Scherk-Schwarz reductions have been described in Ref. [16] and further analyzed in Ref. [25], using the framework of SO$(8,4+m)$ exceptional field theory. We illustrate here the techniques developed in Sec. 3 by computing their Kaluza-Klein spectra.

The group of isometries of six-dimensional supergravity on AdS$_3 \times S^3$ is

$$\text{SO}(2,2) \times \text{SO}(4) \cong \text{SL}(2,\mathbb{R}) \times \text{SL}(2,\mathbb{R}) \times \text{SU}(2) \times \text{SU}(2). \tag{4.1}$$

These isometries are captured within the $\mathcal{N} = 4$ supergroup SU$(2|1,1)$, whose even part is precisely SL$(2,\mathbb{R}) \times$ SU$(2)$ [29]. More precisely, in the case $\mathcal{N}_{6d} = (2,0)$, the relevant supergroup is $\mathcal{G} = \text{SU}(2|1,1)_{\text{L}} \times \text{SU}(2|1,1)_{\text{R}}$, whereas it is $\mathcal{G} = (\text{SL}(2,\mathbb{R}) \times \text{SU}(2))_{\text{L}} \times \text{SU}(2|1,1)_{\text{R}}$ for $\mathcal{N}_{6d} = (1,1)$. The supermultiplets of SU$(2|1,1)$ depend on the representations of two SU$(2)$ factors [17]. The first one is realized as part of the sphere isometries of Eq. (4.1) and is the $R$-symmetry group of SU$(2|1,1)$. It is gauged in three-dimensions, and we will thus denote it SU$(2)_{\text{gauge}}$. The second one is the automorphism group of $\mathfrak{su}(2|1,1)$. It describes a global symmetry of the three-dimensional supergravity, and will thus be noted SU$(2)_{\text{global}}$. The multiplets relevant for our study are the short ones, which we will denote $\boldsymbol{k+1}$. They are given in Tab. 1.

### 4.1.1 $\mathcal{N}_{6d} = (2,0)$

The spectrum around the AdS$_3 \times S^3$ vacuum of $\mathcal{N}_{6d} = (2,0)$ supergravity in six dimensions has been computed in Ref. [5] by standard techniques, *i.e.* linearization of the equations of motion around the AdS$_3$ background. In Ref. [17], group theoretical arguments were used

---

[6]In the case $\mathcal{N}_{6d} = (1,1)$, the tensor multiplet is absorbed into the gravitational multiplet, so that the theory does not feature any coupling.

in the same purpose. The vacuum preserves indeed enough supersymmetries so that one can deduce the entire spectrum, *i.e.* representations and masses, without lengthy calculation. The spectrum is organized under the supergroup $SU(2|1,1)_L \times SU(2|1,1)_R$, whose bosonic extension $SU(2)_{L\,gauge} \times SU(2)_{R\,gauge} \cong SO(4)_{gauge}$ corresponds to the isometry group of the sphere $S^3$, and global factors $SU(2)_{L\,global} \times SU(2)_{R\,global} \cong SO(4)_{global}$ and $SO(m)_{global}$. It consists of a spin-2 and two spin-1 Kaluza-Klein towers, one of them transforming as a vector under $SO(m+1)$. The relevant multiplets are given in Tab. 2, in the notations of Ref. [17]: the spin-1 and spin-2 multiplets, which are scalars under $SO(m+1)$, are noted $[\boldsymbol{k+1}, \boldsymbol{k+1}]_S$ and $[\boldsymbol{p}, \boldsymbol{p+2}]_S + [\boldsymbol{p+2}, \boldsymbol{p}]_S$, respectively. The spin-1 multiplets, which are vectors under $SO(m+1)$, are noted $[\boldsymbol{k+1}, \boldsymbol{k+1}]_S^{(m+1)}$. The full spectrum has been proven to be[7]

$$\mathcal{S}'_{(2,0)} = \sum_{k \geq 2} [\boldsymbol{k+1}, \boldsymbol{k+1}]_S + \sum_{k \geq 1} [\boldsymbol{k+1}, \boldsymbol{k+1}]_S^{(m+1)} + \sum_{p \geq 2} \left( [\boldsymbol{p}, \boldsymbol{p+2}]_S + [\boldsymbol{p+2}, \boldsymbol{p}]_S \right). \quad (4.2)$$

We use this example as a warm up to test the tools we developed in Sec. 3.

To describe the three-dimensional theory, we decomposes the "flat" indices $\bar{M}$ according to

$$\{X^{\bar{M}}\} \longrightarrow \{X^A, X_A, X^\alpha, X^{\hat{\alpha}}\}, \quad (4.3)$$

so that $SO(8, 4+m)$ is decomposed into $GL(4) \times SO(4,m)$. The $GL(4)$ part is embedded into $SO(4,4)$ and the $SO(8, 4+m)$ invariant tensor takes the form

$$\eta_{\bar{M}\bar{N}} = \begin{pmatrix} 0 & \delta_A{}^B & 0 & 0 \\ \delta^B{}_A & 0 & 0 & 0 \\ 0 & 0 & -\delta_{\alpha\beta} & 0 \\ 0 & 0 & 0 & \delta_{\hat{\alpha}\hat{\beta}} \end{pmatrix}. \quad (4.4)$$

The embedding tensor is

$$\theta_{ABCD} = 2\,\varepsilon_{ABCD}, \quad \theta_{ABC}{}^D = \varepsilon_{ABCE}\,\delta^{ED}, \quad (4.5)$$

with all other components vanishing. It induces a gauge group $SO(4)_{gauge} \ltimes T_6$, where $T_6$ denotes an abelian group of six translations transforming in the adjoint representation of $SO(4)_{gauge}$. As shown in Ref. [16], the resulting theory is a consistent truncation that captures the $S^3$ reduction of $\mathcal{N}_{6d} = (2,0)$ six-dimensional supergravity coupled to $m+1$ tensor multiplets. The associated three-dimensional supergravity possesses a $\mathcal{N} = (4,4)$ vacuum at the scalar origin $\mathcal{M}_{\bar{M}\bar{N}} = \delta_{\bar{M}\bar{N}}$.

We construct a complete basis of scalar functions on $S^3$, following closely the construction of Ref. [12]. We consider the elementary round $S^3$ harmonics $\mathcal{Y}^A$, $A \in [\![1,4]\!]$, normalized as $\mathcal{Y}^A \mathcal{Y}^A = 1$. The basis is given by all the polynomials in $\mathcal{Y}^A$:

$$\{\mathcal{Y}^\Sigma\} = \{1, \mathcal{Y}^A, \mathcal{Y}^{A_1 A_2}, \dots, \mathcal{Y}^{A_1 \dots A_n}, \dots\}, \quad (4.6)$$

where we use the notation $\mathcal{Y}^{A_1 \dots A_n} = \mathcal{Y}^{((A_1} \dots \mathcal{Y}^{A_n))}$, with double parenthesis denoting traceless symmetrization. We will denote the integer $n$ as the level of the harmonics tower.

To compute the spectrum, we need the expression of the $\mathcal{T}$ matrices defined in Eq. (3.4). They can be extracted from the generalized Scherk-Schwarz reduction built in Ref. [16, 25]. The twist matrix $U_M{}^{\bar{M}}$ is constructed from the harmonics $\mathcal{Y}^A$ and the round $S^3$ metric

---

[7]The multiplets $[\boldsymbol{2}, \boldsymbol{2}]_S$, $[\boldsymbol{2}, \boldsymbol{4}]_S$ and $[\boldsymbol{4}, \boldsymbol{2}]_S$ can be extracted from Tab. 2 by disregarding the lines with negative SU(2) spins. The massless supergravity multiplets $[\boldsymbol{1}, \boldsymbol{3}]_S$ and $[\boldsymbol{3}, \boldsymbol{1}]_S$ do not carry any degree of freedom in three dimensions and are not included in the spectra.

Tab. 2: Spin-1 $[k+1, k+1]_S$ and spin-2 $[p, p+2]_S$ multiplets of $SU(2|1,1)_L \times SU(2|1,1)_R$, for $k \geq 2$ and $p \geq 3$, constructed from the short multiplets of Tab. 1 [17]. The SO(4) representations are given by a couple of SU(2) spins. The conjugate spin-2 multiplet $[p+2, p]_S$ is obtained by inverting $\Delta_L$ with $\Delta_R$, taking the opposite spin $-s$ and exchanging the SU(2) spins inside each $SO(4)_{\text{gauge}}$ and $SO(4)_{\text{global}}$ representations.

| $\Delta_L$ | $\Delta_R$ | $\Delta$ | $s$ | $SO(4)_{\text{gauge}}$ | $SO(4)_{\text{global}}$ |
|---|---|---|---|---|---|
| | | | **Spin-1 multiplet** $[k+1, k+1]_S$ | | |
| $k/2$ | $k/2$ | $k$ | $0$ | $(k/2, k/2)$ | $(0,0)$ |
| $k/2$ | $(k+1)/2$ | $k+1/2$ | $1/2$ | $(k/2, (k-1)/2)$ | $(0, 1/2)$ |
| $(k+1)/2$ | $k/2$ | $k+1/2$ | $-1/2$ | $((k-1)/2, k/2)$ | $(1/2, 0)$ |
| $(k+1)/2$ | $(k+1)/2$ | $k+1$ | $0$ | $((k-1)/2, (k-1)/2)$ | $(1/2, 1/2)$ |
| $k/2$ | $(k+2)/2$ | $k+1$ | $1$ | $(k/2, (k-2)/2)$ | $(0, 0)$ |
| $(k+2)/2$ | $k/2$ | $k+1$ | $-1$ | $((k-2)/2, k/2)$ | $(0, 0)$ |
| $(k+1)/2$ | $(k+2)/2$ | $k+3/2$ | $1/2$ | $((k-1)/2, (k-2)/2)$ | $(1/2, 0)$ |
| $(k+2)/2$ | $(k+1)/2$ | $k+3/2$ | $-1/2$ | $((k-2)/2, (k-1)/2)$ | $(0, 1/2)$ |
| $(k+2)/2$ | $(k+2)/2$ | $k+2$ | $0$ | $((k-2)/2, (k-2)/2)$ | $(0, 0)$ |
| | | | **Spin-2 multiplet** $[p, p+2]_S$ | | |
| $(p-1)/2$ | $(p+1)/2$ | $p$ | $1$ | $((p-1)/2, (p+1)/2)$ | $(0, 0)$ |
| $(p-1)/2$ | $(p+2)/2$ | $p+1/2$ | $3/2$ | $((p-1)/2, p/2)$ | $(0, 1/2)$ |
| $p/2$ | $(p+1)/2$ | $p+1/2$ | $1/2$ | $((p-2)/2, (p+1)/2)$ | $(1/2, 0)$ |
| $p/2$ | $(p+2)/2$ | $p+1$ | $1$ | $((p-2)/2, p/2)$ | $(1/2, 1/2)$ |
| $(p-1)/2$ | $(p+3)/2$ | $p+1$ | $2$ | $((p-1)/2, (p-1)/2)$ | $(0, 0)$ |
| $(p+1)/2$ | $(p+1)/2$ | $p+1$ | $0$ | $((p-3)/2, (p+1)/2)$ | $(0, 0)$ |
| $p/2$ | $(p+3)/2$ | $p+3/2$ | $3/2$ | $((p-2)/2, (p-1)/2)$ | $(1/2, 0)$ |
| $(p+1)/2$ | $(p+2)/2$ | $p+3/2$ | $1/2$ | $((p-3)/2, p/2)$ | $(0, 1/2)$ |
| $(p+1)/2$ | $(p+3)/2$ | $p+2$ | $1$ | $((p-3)/2, (p-1)/2)$ | $(0, 0)$ |

$h_{ij} = \partial_i \mathcal{Y}^A \partial_j \mathcal{Y}^A$, where $\partial_i$ denotes the partial derivative with respect to the physical internal coordinates $\{y^i\}$, $i \in [\![1, 3]\!]$ properly embedded into $\{Y^{MN}\}$. The weight factor is defined by $\rho = h^{-1/2}$. The only non-vanishing components of the operator in Eq. (3.4) are

$$\rho^{-1} U^M{}_A U^N{}_B \partial_{MN} = \varepsilon_{ABCD} K^{CD\,i} \partial_i, \tag{4.7}$$

with the round $S^3$ Killing vectors $K^{AB\,i} = h^{ij} \partial_j \mathcal{Y}^{[A} \mathcal{Y}^{B]}$. Following Eq. (4.6) and (4.7), the matrices $\mathcal{T}_{\bar{M}\bar{N}}$ are block-diagonal level by level and each block has the form

$$\mathcal{T}_{\bar{M}\bar{N}}{}^{A_1 \dots A_n B_1 \dots B_n} = n \mathcal{T}_{\bar{M}\bar{N}\,((A_1}{}^{((B_1} \delta_{A_2}{}^{B_2} \dots \delta_{A_n))}{}^{B_n))}, \tag{4.8}$$

where we lowered the indices $A_i$ in the right-hand side for readability. The level 1 block $\mathcal{T}_{\bar{M}\bar{N}}{}^{AB}$

is finally given by its only non vanishing components

$$\mathcal{T}_{CD}{}^{AB} = -\varepsilon_{CDEF}\,\delta^{AE}\delta^{BF}\,. \tag{4.9}$$

We then have all the information needed to compute the mass matrices of Sec. 3. Injecting the embedding tensor (4.5) and the $\mathcal{T}$ matrices (4.8) in the mass matrices (3.6), (3.17) and (3.23), we extract the mass eigenvalues of the bosonic degrees of freedom. We then deduce the conformal dimensions (3.26) and compute the weights $\Delta_{\mathrm{L,R}}$ knowing the spins $s$. We can then infer the fermionic masses from the structure of the multiplets of Tab. 2. At levels 0 and $n \geq 1$, fields combine into the multiplets

$$\begin{cases} \mathcal{S}^{(0)}_{(2,0)} = [\mathbf{3},\mathbf{3}]_S + [\mathbf{2},\mathbf{2}]^{(m)}_S, \\ \mathcal{S}^{(n)}_{(2,0)} = [\mathbf{n+1},\mathbf{n+1}]_S + [\mathbf{n+3},\mathbf{n+3}]_S + [\mathbf{n+2},\mathbf{n+2}]^{(m)}_S \\ \qquad + [\mathbf{n+1},\mathbf{n+3}]_S + [\mathbf{n+3},\mathbf{n+1}]_S\,. \end{cases} \tag{4.10}$$

The full spectrum is then

$$\mathcal{S}_{(2,0)} = \sum_{n \geq 0} \mathcal{S}^{(n)}_{(2,0)}, \tag{4.11}$$

which precisely coincide with Eq. (4.2). Our method thus successfully reproduces the computations of Ref. [5, 17], and allows to bypass standard harmonic analysis.

### 4.1.2 $\mathcal{N}_{\mathrm{6d}} = (\mathbf{1}, \mathbf{1})$

Let us now turn to the compactification of $\mathcal{N}_{\mathrm{6d}} = (1,1)$ six-dimensional supergravity on $\mathrm{AdS}_3 \times S^3$. As mentioned above, the vacuum is only quarter-maximal and it preserves $\mathcal{N} = (0,4)$ supersymmetries in three dimensions. Contrary to the previous example, the spectrum is not sufficiently constrained by supersymmetry and cannot be computed using group theory. One can however use group theory to compute the bosonic representations that appear in the spectrum, and finds that they formally combine into $\mathcal{N} = (4,4)$ multiplets of $\mathrm{SU}(2|1,1)_{\mathrm{L}} \times \mathrm{SU}(2|1,1)_{\mathrm{R}}$. According to Ref. [17], it yields[8]

$$\mathcal{S}'_{(1,1)} = [\mathbf{2},\mathbf{2}]_S + 2\sum_{k \geq 2}[\mathbf{k+1},\mathbf{k+1}]_S + \sum_{k \geq 1}[\mathbf{k+1},\mathbf{k+1}]^{(m)}_S + \sum_{p \geq 2}\Big([\mathbf{p},\mathbf{p+2}]_S + [\mathbf{p+2},\mathbf{p}]_S\Big). \tag{4.12}$$

However, the vacuum has only $\mathcal{N} = (0,4)$ supersymmetries, so that only the factor $\mathrm{SU}(2|1,1)_{\mathrm{R}}$ is preserved and the conformal dimensions assigned by Eq. (4.12) cannot be trusted. From $\mathrm{SU}(2|1,1)_{\mathrm{L}}$ survives only the even part $\mathrm{SL}(2,\mathbb{R})_{\mathrm{L}} \times \mathrm{SU}(2)_{\mathrm{L\,gauge}}$ and a global $\mathrm{SU}(2)_{\mathrm{L\,global}}$. The relevant multiplets are thus the short ones of $\mathrm{SU}(2|1,1)_{\mathrm{R}}$, given in Tab. 1, associated to a representation of $\mathrm{SU}(2)_{\mathrm{L\,global}} \times \mathrm{SU}(2)_{\mathrm{L\,gauge}}$ with conformal dimension $\Delta_{\mathrm{L}}$. We will then use the notation $(\mathbf{k+1})^{(j_{\mathrm{gl}}, j_{\mathrm{ga}})}_{\Delta_{\mathrm{L}}}$ to denote those multiplets, with $j_{\mathrm{gl}}$ and $j_{\mathrm{ga}}$ spins of $\mathrm{SU}(2)_{\mathrm{L\,global}}$ and $\mathrm{SU}(2)_{\mathrm{L\,gauge}}$, respectively, and add an exponent $m$ for multiplets transforming as vectors of $\mathrm{SO}(m)$. In these notations, the spectrum of Eq. (4.12) decomposes into $\mathrm{SU}(2|1,1)_{\mathrm{R}}$ multiplets

---

[8]The range of the sum is not explicit in Ref. [17] and requires further analysis.

as

$$\mathcal{S}'_{(1,1)} = 2^{(0,1/2)}_{1/2} + 2^{(1/2,0)}_{1}$$
$$+ 2\sum_{k\geq 2}\left[(k+1)^{(0,k/2)}_{k/2} + (k+1)^{(1/2,(k-1)/2)}_{(k+1)/2} + (k+1)^{(0,(k-2)/2)}_{(k+2)/2}\right]$$
$$+ \sum_{k\geq 1}\left[(k+1)^{(0,k/2),m}_{k/2} + (k+1)^{(1/2,(k-1)/2),m}_{(k+1)/2} + (k+1)^{(0,(k-2)/2),m}_{(k+2)/2}\right] \qquad (4.13)$$
$$+ \sum_{p\geq 2}\Big[(p+2)^{(0,(p-1)/2)}_{(p-1)/2} + (p+2)^{(1/2,(p-2)/2)}_{p/2} + (p+2)^{(0,(p-3)/2)}_{(p+1)/2}$$
$$+ p^{(0,(p+1)/2)}_{(p+1)/2} + p^{(1/2,p/2)}_{(p+2)/2} + p^{(0,(p-1)/2)}_{(p+3)/2}\Big].$$

As the factor $SU(2|1,1)_L$ is not preserved, the conformal dimensions $\Delta_L$ are not restricted to the values given in Tab. 2 and could in fact be different from those predicted in the spectrum (4.13). As our tools allow to compute the spectrum around vacua preserving few, or no, supersymmetries, we use them in the following to adjust the masses in Eq. (4.13).

We use the same index split (4.3) as in the previous example. The theory in three dimensions is described by the following embedding tensor:

$$\theta_{AB} = 4\,\delta_{AB}, \quad \theta_{ABCD} = 2\,\varepsilon_{ABCD}, \qquad (4.14)$$

and all other components vanish. The associated gauge group is $SO(4)_{\text{gauge}} \ltimes \left(T_6 \times (T_4)^{4+m}\right)$, where $T_4$ and $T_6$ denote abelian groups of 4 and 6 translations transforming in the vectorial and adjoint representations of $SO(4)_{\text{gauge}}$, respectively. The associated theory is a consistent truncation of $\mathcal{N}_{6d} = (1,1)$ supergravity in six dimensions coupled to $m$ vector multiplets on the round $S^3$ [16]. Its potential has an $AdS_3$ vacuum at the scalar origin $\mathcal{M}_{\bar{M}\bar{N}} = \delta_{\bar{M}\bar{N}}$ that preserves $\mathcal{N} = (0,4)$ supersymmetries.

The generalized Scherk-Schwarz reduction leading to Eq. (4.14) has been described in Ref. [16] with the same geometrical data as in the previous section. The action of the operator in Eq. (3.4) is now given by[9]

$$\rho^{-1}U^M{}_A U^N{}_B \partial_{MN} = 2K_{AB}{}^i\partial_i, \qquad (4.15)$$

with the Killing vectors introduced in Eq. (4.7)[10] and all other components vanishing. We then use the same basis (4.6) of scalar functions on $S^3$ as previously, so that the expression (4.8) of the matrices $\mathcal{T}_{\bar{M}\bar{N}}$ is still valid, however with the level 1 block $\mathcal{T}_{\bar{M}\bar{N}}{}^{AB}$ defined as

$$\mathcal{T}_{CD}{}^{AB} = -2\,\delta_{[C}{}^A\delta_{D]}{}^B \qquad (4.16)$$

and all other components vanishing.

Again, we combine the expressions of the embedding tensor (4.14) and of the $\mathcal{T}$ matrices with the mass matrices (3.6), (3.17) and (3.23) to compute mass eigenvalues of the spin-2, vector and scalar fields. We use Eq. (3.26) to define their conformal dimensions. We then deduce the full spectrum knowing the multiplets in which the fields should lie. At levels 0, 1

---

[9]Contrary to Ref. [16,25], we embedded the physical internal coordinates as $\partial_i = -\partial_{0i}$, in agreement with our normalizations.

[10]The indices $A, B$ are lowered using the identity matrix $\delta_{AB}$.

and $n \geq 2$, fields arrange into the multiplets

$$
\begin{cases}
\mathcal{S}_{(1,1)}^{(0)} = \mathbf{2}_{3/2}^{(0,1/2),m} + \mathbf{2}_{1}^{(1/2,0),m} + \mathbf{3}_{1}^{(0,1)} + \mathbf{3}_{1/2}^{(1/2,1/2)} + \mathbf{3}_{2}^{(0,0)}, \\
\mathcal{S}_{(1,1)}^{(1)} = \mathbf{2}_{1/2}^{(0,1/2)} + \mathbf{2}_{1}^{(1/2,1)} + \mathbf{2}_{3/2}^{(0,3/2)} + \mathbf{2}_{2}^{(1/2,0)} + \mathbf{2}_{5/2}^{(0,1/2)} + \mathbf{3}_{2}^{(0,1),m} + \mathbf{3}_{3/2}^{(1/2,1/2),m} + \mathbf{3}_{1}^{(0,0),m} \\
\qquad + \mathbf{4}_{1/2}^{(0,1/2)} + \mathbf{4}_{1}^{(1/2,1)} + \mathbf{4}_{3/2}^{(0,3/2)} + \mathbf{4}_{2}^{(1/2,0)} + \mathbf{4}_{5/2}^{(0,1/2)}, \\
\mathcal{S}_{(1,1)}^{(n)} = (n+1)_{n/2}^{(0,n/2)} + (n+3)_{n/2}^{(0,n/2)} + (n+1)_{(n+1)/2}^{(1/2,(n+1)/2)} + (n+3)_{(n+1)/2}^{(1/2,(n+1)/2)} \\
\qquad + (n+1)_{(n+2)/2}^{(0,(n-2)/2)} + (n+3)_{(n+2)/2}^{(0,(n-2)/2)} + (n+1)_{(n+2)/2}^{(0,(n+2)/2)} + (n+3)_{(n+2)/2}^{(0,(n+2)/2)} \\
\qquad + (n+1)_{(n+3)/2}^{(1/2,(n-1)/2)} + (n+3)_{(n+3)/2}^{(1/2,(n-1)/2)} + (n+1)_{(n+4)/2}^{(0,n/2)} + (n+3)_{(n+4)/2}^{(0,n/2)} \\
\qquad + (n+2)_{(n+3)/2}^{(0,(n+1)/2),m} + (n+2)_{(n+2)/2}^{(1/2,n/2),m} + (n+2)_{(n+1)/2}^{(0,(n-1)/2),m} \Big].
\end{cases}
\tag{4.17}
$$

Adding all the levels, we get the full spectrum

$$
\begin{aligned}
\mathcal{S}_{(1,1)} = {}& \mathbf{2}_{1/2}^{(0,1/2)} + \mathbf{2}_{2}^{(1/2,0)} \\
& + \sum_{k \geq 2} \Big[ (k+1)_{k/2}^{(0,k/2)} + (k+1)_{(k+3)/2}^{(1/2,(k-1)/2)} + (k+1)_{(k+2)/2}^{(0,(k-2)/2)} \\
& \qquad + (k+1)_{k/2}^{(0,k/2)} + (k+1)_{(k-1)/2}^{(1/2,(k-1)/2)} + (k+1)_{(k+2)/2}^{(0,(k-2)/2)} \Big] \\
& + \sum_{k \geq 1} \Big[ (k+1)_{(k+2)/2}^{(0,k/2),m} + (k+1)_{(k+1)/2}^{(1/2,(k-1)/2),m} + (k+1)_{k/2}^{(0,(k-2)/2),m} \Big] \\
& + \sum_{p \geq 2} \Big[ (p+2)_{(p-1)/2}^{(0,(p-1)/2)} + (p+2)_{(p+2)/2}^{(1/2,(p-2)/2)} + (p+2)_{(p+1)/2}^{(0,(p-3)/2)} \\
& \qquad + p_{(p+1)/2}^{(0,(p+1)/2)} + p_{p/2}^{(1/2,p/2)} + p_{(p+3)/2}^{(0,(p-1)/2)} \Big].
\end{aligned}
\tag{4.18}
$$

This coincides with the spectrum of Eq. (4.13) from the point of view of the SU(2) representations, but the multiplets differ: as expected from the supersymmetry breaking from $\mathcal{N} = (4,4)$ to $\mathcal{N} = (0,4)$, the weights $\Delta_{\mathrm{L}}$ are not the ones of SU(2|1,1)$_{\mathrm{L}}$ multiplets. The spectrum (4.18) thus organizes into genuine $\mathcal{N} = (0,4)$ multiplets, and cannot be recombined into $\mathcal{N} = (4,4)$ ones. Thus, $\mathcal{S}_{(1,1)}$ describes the entire spectrum of $\mathcal{N}_{6d} = (1,1)$ six-dimensional supergravity on AdS$_3 \times S^3$, with representations and masses.

## 4.2 Ten-dimensional supergravity on AdS$_3 \times S^3 \times S^3 \times S^1$

We now turn to the spectrum of ten-dimensional maximal supergravity on AdS$_3 \times S^3 \times S^3 \times S^1$, whose vacuum preserves only half of the supersymmetries. The group of isometries is given by two copies of the large $\mathcal{N} = 4$ supergroup: $\mathcal{G} = D^1(2,1;\alpha)_{\mathrm{L}} \times D^1(2,1;\alpha)_{\mathrm{R}}$, with $\alpha$ the ratio of the spheres $S^3$ radii.

Even if half of the supersymmetries are preserved at the vacuum, supersymmetry does not constrain the spectrum sufficiently to allow its computation using representation theory only. As pointed out in Ref. [18], the Kaluza-Klein states fall into short multiplets of $D^1(2,1;\alpha) \times D^1(2,1;\alpha)$, most of which could be combined to form long multiplets. As the conformal dimensions of the long representations are not fixed, group theory fails to predict the masses that appear in the spectrum. In Ref. [19], the scalar masses around the AdS$_3$ vacuum have been computed by standard analysis, and further used to infer the entire Kaluza-Klein spectrum of the theory. It confirmed that indeed most of the fields arrange in long representations.

We compute here the masses of all the bosonic fields around the vacuum. As our tools apply to half-maximal supergravity, we consider the truncation to $\mathcal{N}_{10d} = 1$ supergravity and we will

reproduce only a subsector of the spectrum. It turns out that, similarly to the construction in Sec. 4.1.2, the vacuum in this truncation breaks another half of the supersymmetries and gives rise to an $\mathcal{N} = (0,4)$ vacuum in three dimensions. Accordingly, only the factor $D^1(2,1;\alpha)_R$ is preserved and $D^1(2,1;\alpha)_L$ is broken to its even part. The even part of $D^1(2,1;\alpha)$ is isomorphic to $SL(2,\mathbb{R}) \times SO(3)^+ \times SO(3)^-$ [29], so that the bosonic symmetries at the vacuum are given by

$$SL(2,\mathbb{R})_L \times SO(3)_L^+ \times SO(3)_L^- \times \underbrace{SL(2,\mathbb{R})_R \times SO(3)_R^+ \times SO(3)_R^-}_{\subset D^1(2,1;\alpha)_R}. \tag{4.19}$$

As in Sec. 4.1, the $SL(2,\mathbb{R})_L \times SL(2,\mathbb{R})_R$ factors combine into the $AdS_3$ isometry group $SO(2,2)$, and the compact ones $SO(3)_L^+ \times SO(3)_R^+ \times SO(3)_L^- \times SO(3)_R^-$ build the isometry groups $SO(4)^\pm = SO(3)_L^\pm \times SO(3)_R^\pm$ of the two spheres, which we denote by $S^{3\pm}$. The three-dimensional theory then features $SO(4)^+ \times SO(4)^-$ as a gauge group and the scalars form the coset space $SO(8,8)/(SO(8) \times SO(8))$.

We need to build an appropriate three-dimensional theory that is a consistent truncation from ten dimensions on $S^{3+} \times S^{3-} \times S^1$. Let's first consider the reduction on $S^{3+} \times S^{3-}$ to four dimensions, with isometry group $SO(4)^+ \times SO(4)^-$. The generic construction of consistent truncations on an internal space of isometry group $G \times G$, with $G$ a Lie group of dimension $d$, has been considered in Ref. [32] using double field theory. It results in a low-dimensional theory carrying gauge fields, a two-form and scalar fields parameterizing the coset space $SO(d,d)/(SO(d) \times SO(d))$. The construction is not specific to three dimensions, so that the embedding tensor do not take the form (2.15) but rather the generic expression $F_{mn}{}^p$, with $SO(d,d)$ indices $m,n,p \in [\![1,2d]\!]$. We are specifically interested in this construction for $G = SO(4)$. Splitting the $SO(6,6)$ indices $m$ according to

$$\{X^m\} \longrightarrow \{X^i, X^{\hat{\imath}}, X^r, X^{\hat{r}}\}, \tag{4.20}$$

with $i, \hat{\imath}, r, \hat{r} \in [\![1,3]\!]$ and writing the $SO(6,6)$ invariant tensor as

$$\eta_{mn} = \begin{pmatrix} -\delta_{ij} & 0 & 0 & 0 \\ 0 & -\delta_{\hat{\imath}\hat{\jmath}} & 0 & 0 \\ 0 & 0 & \delta_{rs} & 0 \\ 0 & 0 & 0 & \delta_{\hat{r}\hat{s}} \end{pmatrix}, \tag{4.21}$$

the embedding tensor $F_{mnp} = F_{mn}{}^q \eta_{pq}$ takes the form

$$\begin{cases} F_{ijk} = \varepsilon_{ijk}, \\ F_{\hat{\imath}\hat{\jmath}\hat{k}} = \alpha \, \varepsilon_{\hat{\imath}\hat{\jmath}\hat{k}}, \end{cases} \qquad \begin{cases} F_{rst} = -\varepsilon_{rst}, \\ F_{\hat{r}\hat{s}\hat{t}} = -\alpha \, \varepsilon_{\hat{r}\hat{s}\hat{t}}, \end{cases} \tag{4.22}$$

and all other components vanishing. Eq. (4.22) shows that $\alpha$ is the relative coupling constant between the isometry groups of the two spheres.

We further compactify on a circle $S^1$ down to three dimensions, where the scalar coset is enhanced to $SO(7,7)/(SO(7) \times SO(7))$. The embedding tensor (4.22) induces a potential that does not admit any AdS stationary point [32]. However, in the same spirit as what has been done in Ref. [31] for the reduction of six-dimensional supergravity on $S^3$, we can take advantage in the fact that the low-dimensional theory lives in three dimensions to stabilize the potential. In three dimensions, the two-form is auxiliary and can be integrated out. It gives rise to an enhanced scalar coset $SO(8,8)/(SO(8) \times SO(8))$ and an additional contribution to the scalar potential, which can be tuned to give rise to a stationary $AdS_3$ point.

The $SO(8,8)$ flat indices $\bar{M}$ are split according to

$$\{X^{\bar{M}}\} \longrightarrow \{X^m, X^+, X^{\hat{+}}, X^-, X^{\hat{-}}\}, \tag{4.23}$$

and the associated invariant tensor is

$$
\eta_{\tilde{M}\tilde{N}} = \begin{pmatrix} \eta_{mn} & 0 & 0 \\ 0 & 0 & \mathbb{1}_2 \\ 0 & \mathbb{1}_2 & 0 \end{pmatrix},
\tag{4.24}
$$

with $\mathbb{1}_2$ the $2 \times 2$ identity matrix. We then construct the three-dimensional embedding tensor $X_{\tilde{M}\tilde{N}|\tilde{P}\tilde{Q}}$ using $F_{mnp}$, and adding a component $\xi$ associated to the integration of the two-form:

$$
\theta_{mnp+} = F_{mnp}, \quad \theta_{++} = \xi.
\tag{4.25}
$$

The potential is stabilized at the scalar origin $\mathcal{M}_{\tilde{M}\tilde{N}} = \delta_{\tilde{M}\tilde{N}}$ if $\xi = 4\sqrt{2}\sqrt{1+\alpha^2}$, and it then takes the value $V_0 = -(1+\alpha^2)/2$. The spacetime at the vacuum is AdS$_3$ and only half of the supersymmetries are preserved: $\mathcal{N} = (0,4)$. The gauge group is $\left(\text{SO}(4)^+ \ltimes (\text{T}_3 \times \text{T}_3)\right) \times \left(\text{SO}(4)^- \ltimes (\text{T}_3 \times \text{T}_3)\right) \times (\text{T}_1)^2$, where $\text{T}_3$ denotes an abelian group of three translations transforming in the vectorial representation of SO(3), and $\text{T}_1$ stands for a translation singlet under $\text{SO}(4)^+ \times \text{SO}(4)^-$.

We now turn to the definition of suitable $\mathcal{T}_{\tilde{M}\tilde{N}}$ matrices. In the previous examples, we used explicit constructions of twist matrices to define $\mathcal{T}_{\tilde{M}\tilde{N}}$. We can in fact bypass the construction of a twist matrix by imposing the condition that the matrices $\mathcal{T}_{\tilde{M}\tilde{N}}$ should correspond to the generators of $\text{SO}(4)^+ \times \text{SO}(4)^-$ in the representation of the chosen scalar harmonics, normalized as in Eq. (3.5). We then consider two sets of SO(4) harmonics $\{\mathcal{Y}^{\dot{A}}\}_{\dot{A}\in[\![1,4]\!]}$ and $\{\mathcal{Y}^{\hat{A}}\}_{\hat{A}\in[\![1,4]\!]}$, defined as functions of the internal physical coordinates $\{y^{\dot{\alpha}}\}_{\dot{\alpha}\in[\![1,3]\!]}$ and $\{y^{\hat{\alpha}}\}_{\hat{\alpha}\in[\![1,3]\!]}$ respectively, and form the $\text{SO}(4)^+ \times \text{SO}(4)^-$ scalar harmonics $\{\mathcal{Y}^A\} = \{\mathcal{Y}^{\dot{A}}, \mathcal{Y}^{\hat{A}}\}$, $A \in [\![1,8]\!]$, which depends on the physical coordinates $\{y^{\alpha}\} = \{y^{\dot{\alpha}}, y^{\hat{\alpha}}\}$, $\alpha \in [\![1,6]\!]$, and are normalized as $\mathcal{Y}^A \mathcal{Y}^A = 1$. We still use Eq. (4.6) to define the full basis of scalar functions. With this parametrization, we again take profit of the expression (4.8) of the matrices $\mathcal{T}_{\tilde{M}\tilde{N}}$, which allows to build the level one matrices $\mathcal{T}_{\tilde{M}\tilde{N}}{}^{AB}$ only. Given Eq. (4.22), we define

$$
\begin{cases}
\mathcal{T}_i{}^{\dot{A}\dot{B}} = \delta_i{}^{[\dot{A}}\delta_4{}^{\dot{B}]} + \dfrac{1}{2}\varepsilon_{i\,4\,\dot{C}\dot{D}}\,\delta_{\dot{C}}{}^{[\dot{A}}\delta_{\dot{D}}{}^{\dot{B}]}, \\[2mm]
T_r{}^{\dot{A}\dot{B}} = -\delta_r{}^{[\dot{A}}\delta_4{}^{\dot{B}]} + \dfrac{1}{2}\varepsilon_{r\,4\,\dot{C}\dot{D}}\,\delta_{\dot{C}}{}^{[\dot{A}}\delta_{\dot{D}}{}^{\dot{B}]},
\end{cases}
\tag{4.26}
$$

so that

$$
\begin{cases}
[\mathcal{T}_i, \mathcal{T}_j]^{\dot{A}\dot{B}} = -\varepsilon_{ijk}\,\mathcal{T}_k{}^{\dot{A}\dot{B}}, \\[2mm]
[\mathcal{T}_r, \mathcal{T}_s]^{\dot{A}\dot{B}} = -\varepsilon_{rst}\,\mathcal{T}_t{}^{\dot{A}\dot{B}}.
\end{cases}
\tag{4.27}
$$

We define accordingly $\mathcal{T}_{\hat{i}}{}^{\hat{A}\hat{B}}$ and $\mathcal{T}_{\hat{r}}{}^{\hat{A}\hat{B}}$ by adding a global factor $\alpha$ and changing all $\dot{A}, \dot{B}$ to $\hat{A}, \hat{B}$ in Eq. (4.26). Finally, we embed these matrices in $\mathcal{T}_{\tilde{M}\tilde{N}}{}^{AB}$ as follows:

$$
\begin{cases}
\mathcal{T}_{i+}{}^{\dot{A}\dot{B}} = \mathcal{T}_i{}^{\dot{A}\dot{B}}, \\[2mm]
\mathcal{T}_{\hat{i}+}{}^{\hat{A}\hat{B}} = \mathcal{T}_{\hat{i}}{}^{\hat{A}\hat{B}},
\end{cases}
\qquad
\begin{cases}
\mathcal{T}_{r+}{}^{\dot{A}\dot{B}} = \mathcal{T}_r{}^{\dot{A}\dot{B}}, \\[2mm]
\mathcal{T}_{\hat{r}+}{}^{\hat{A}\hat{B}} = \mathcal{T}_{\hat{r}}{}^{\hat{A}\hat{B}}.
\end{cases}
\tag{4.28}
$$

Together with Eq. (4.22), (4.25) and (4.27), this definition ensures that Eq. (3.5) is satisfied, assuring that the matrices $\mathcal{T}_{\tilde{M}\tilde{N}}$ generate $\text{SO}(4)^+ \times \text{SO}(4)^-$ with the appropriate normalization.

We finally put the expressions (4.25) and (4.28) into the mass formulas of Sec. 3 and compute the mass eigenvalues. The spectrum organizes into representations of $D^1(2,1;\alpha)$, which are labeled by two half integer parameters $(\ell^+, \ell^-)$[11] [18]. With our construction, the representations $(\ell^+, \ell^-)$ appearing at level $n$ satisfy

$$
\ell^+ + \ell^- = \frac{n}{2}.
\tag{4.29}
$$

---

[11]The $\ell^{\pm}$ in our conventions correspond to the $j^{\pm}$ of Ref. [19].

We obtain scalar masses that feature a highly non trivial dependence on $\alpha$:[12]

$$
\begin{cases}
\left(m_{\ell^+,\ell^-}\,\ell_{\mathrm{AdS}}\right)^2 = \dfrac{4}{1+\alpha^2}\left(\ell^+\left(\ell^++1\right)+\alpha^2\,\ell^-\left(\ell^-+1\right)\right), \\[2mm]
\left(m^{(\pm)}_{\ell^+,\ell^-}\,\ell_{\mathrm{AdS}}\right)^2 = -1 + \left(2\pm\sqrt{1+\dfrac{4}{1+\alpha^2}\left(\ell^+(\ell^++1)+\alpha^2\,\ell^-(\ell^-+1)\right)}\,\right)^2 .
\end{cases}
\tag{4.30}
$$

The masses of the vector and the spin-2 fields accordingly complete the associated $D^1(2,1;\alpha)$ long representations. The expressions of the scalar masses in Eq. (4.30) reproduce exactly the ones computed in Ref. [19]. Our construction allows to bypass lengthy calculations and extend the analysis to the spin-1 and spin-2 sectors. As we describe a vacuum of the half-maximal theory, the constructed theory cannot reproduce the full $D^1(2,1;\alpha)_{\mathrm{L}}\times D^1(2,1;\alpha)_{\mathrm{R}}$ spectrum. It reproduces however a subsector thereof. This subsector together with supersymmetry is sufficient to deduce the entire spectrum of the maximal theory in terms of long multiplets of $D^1(2,1;\alpha)_{\mathrm{L}}\times D^1(2,1;\alpha)_{\mathrm{R}}$.

Our analysis can be extended to the maximal theory. The truncation described by the embedding tensor (4.25) properly embedded into $\mathrm{E}_{8(8)}$ exceptional field theory [10] is indeed consistent by construction, and leads to the maximal three-dimensional supergravity constructed in Ref. [34]. It shows that this theory is a consistent truncation. Extending our mass formulas of Sec. 3 to the full $\mathrm{E}_{8(8)}$ exceptional field theory would then explicitly reproduce the complete mass spectrum.

# 5 Conclusion

In this paper, we developed tools to compute the bosonic Kaluza-Klein spectrum around any vacuum of a half-maximal gauged supergravity in three dimensions that arises from a consistent truncation of higher-dimensional supergravity. To do so, we used the framework of $\mathrm{SO}(8,p)$ exceptional field theory. This is an extension of the techniques developed in Ref. [11, 12], which focused on reduction to maximal gauged supergravity in four and five dimensions. Our main results are the mass matrices (3.6), (3.17) and (3.23) for spin-2, vector and scalar fields, respectively. They are expressed in terms of an embedding tensor, which describes the three-dimensional supergravity, and of so-called $\mathcal{T}$ matrices, which encode the linear action on scalar harmonics associated to the compactification.

We have illustrated the efficiency of the method by compactly reproducing the spectrum of six-dimensional $\mathcal{N}=(2,0)$ supergravity on $\mathrm{AdS}_3\times S^3$, originally computed in Ref. [5,17], and the highly non-trivial masses of the $\mathrm{AdS}_3\times S^3\times S^3$ vacuum computed in Ref. [19], which are organized into multiplets of the supergroup $D^1(2,1;\alpha)$. We also derived the spectrum of six-dimensional $\mathcal{N}=(1,1)$ supergravity on $\mathrm{AdS}_3\times S^3$ and corrected the predictions of Ref. [17].

In particular, the technique makes it possible to compute the spectra around vacua with few or no remaining symmetries [12,35], as illustrated in Ref. [36] in the case of the non-supersymmetric $\mathrm{SO}(3)\times\mathrm{SO}(3)$-invariant $\mathrm{AdS}_4$ vacuum of eleven-dimensional supergravity. Though the lowest modes of the consistent truncation to four dimensions are above the Breitenlohner-Freedman bound [37,38], the higher Kaluza-Klein modes are tachyonic so that the vacuum is perturbatively unstable. Similar techniques were applied in Ref. [39] to prove the pertubative stability of the Kaluza-Klein spectrum around the $\mathrm{G}_2$-invariant non-supersymmetric $\mathrm{AdS}_4$ solution of massive IIA supergravity. The question of the stability of non-supersymmetric vacua may also be asked in three dimensions. For example, there exists

---

[12]Analogous masses have been obtained in Ref. [33] in the spectrum of the Laplacian operator on the three-dimensional Heisenberg nilmanifold.

a one-parameter family of non-supersymmetric vacua within the $\mathcal{N} = (2,0)$ and $\mathcal{N} = (1,1)$ AdS$_3 \times S^3$ theories [25]. There is an interval of the parameter within which the lowest modes of the spectra are stable. The parametrization of Sec. 4.1 allows to compute the whole Kaluza-Klein tower around these vacua. In the light of the AdS swampland conjecture [40], which speculates that all non-supersymmetric AdS vacua within string theory are unstable, it will be very interesting to know whether the stability survives at all levels. We hope to report on this soon. This method may also find application in streamlining and extending the analysis of unstable AdS$_3$ vacua such as Ref. [41,42] and study the possibilities to get rid of those modes through appropriate projections.

One additional advantage of the method is to provide access to the origin of the mass eigenstates in the higher-dimensional theory. The method does not only provide the mass eigenvalues, it also provides the associated eigenvectors in the variables of exceptional field theory. We can then translate them back into the original higher-dimensional variables using the explicit dictionary relating the exceptional field theory fields with the higher dimensional supergravity. Such a dictionary has been established in Ref. [25] for the examples of Sec. 4.1.

The possibility to efficiently compute Kaluza-Klein spectra around AdS vacua is also a key tool in the context of the AdS/CFT correspondence. The masses of the Kaluza-Klein modes encodes the conformal dimensions of operators in the dual theory, which often cannot be computed directly, except for protected operators. The knowledge of the whole spectrum can also be used as a test of the duality, as has been done in Ref. [43,44] in the context of string theory on AdS$_3 \times S^3 \times S^3 \times S^1$. A similar analysis could *e.g.* be conducted for the $\mathcal{N} = (0,4)$ solutions of massive type IIA supergravity with AdS$_3 \times S^2$ factors exhibited in Ref. [45,46].

Another path to be explored is the use of the developed tools to infer if a given AdS$_3$ vacuum of three-dimensional gravity could be embedded as a consistent truncation into higher-dimensional supergravities. Indeed, since the mass formulas do not require an explicit twist matrix, we can extract information on the possible truncation using the three-dimensional theory only. It will be interesting to apply these ideas to the AdS$_3$ vacua constructed in Ref. [47].

It will finally be relevant to extend the formalism to maximal three dimensional supergravity using E$_{8(8)}$ exceptional field theory [10]. The maximal SO(8) × SO(8) gauged theory of Ref. [48] admits a large amount of vacua, and at least a non-supersymmetric one with stable lowest level [49], whose higher Kaluza-Klein modes stability could be examined. An extension of the method to the maximal case will also allow to extend the construction of Sec. 4.2 for the AdS$_3 \times S^3 \times S^3 \times S^1$ vacuum and to identify the entire $D^1(2,1;\alpha) \times D^1(2,1;\alpha)$ spectrum.

## Acknowledgment

It is a pleasure to thank Henning Samtleben, for guiding me through this project and carefully reading the manuscript. I also would like to thank Gabriel Larios for his useful advices and fruitful discussions.

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
