# Peer review of "Kaluza-Klein spectrometry for ${\rm AdS_{3}}$ vacua"

_SciPost Physics, doi:SciPost Phys. 10, 131 (2021)_

## Round 1 · Referee Report · Anonymous (Referee 1) · 2020-12-28

Report

This paper formulates a very effective technique to compute the mass spectra of Kaluza-Klein fluctuations around certain classes of supergravity solutions by exploiting a reformulation of half-maximal supergravities in terms of $D=3$ extended field theory based on an O$(8,n)$ duality group. These solutions are products of an "external" AdS$_3$/Minkoswski$_3$/dS$_3$ factor with an internal generalised parallelisable space (which gives rise to a so-called generalised Scherk-Schwarz reduction to a $D=3$ gauged supergravity). This class of solutions includes some known and previously studied AdS$_3$ solutions which the author uses to exemplify the effectiveness of the formalism.

This whole approach is a generalisation of the results in 1911.12640 and 2009.03347, which apply to backgrounds with an "external" factor of dimension $D\ge4$. There are subtleties inherent in the extended generalised geometry for the $D=3$ case as well as in the structure of the equations of motion that make the analysis in this paper a non-trivial generalisation of the previous literature and certainly worth the effort, given how it greatly simplifies the study of certain Kaluza--Klein spectra.

The paper is well written and detailed and will certainly provide a basis for future studies of mass spectra for much larger classes of AdS$_3$ supergravity solutions in the future, and their application to AdS/CFT and to the swampland program. I believe the following point needs to be addressed before publication:

1) The Ansatz for the Kaluza-Klein fluctuations of the constrained vector fields $\mathcal{B}_{\mu\,MN}$ in equation (3.2) should be motivated explicitly, given that these fields are one of the main differences between $D=3$ and higher-$D$ extended field theories. I can see how the expression provided is the natural generalisation of the generalised Scherk-Schwarz Ansatz (2.13), and that such natural generalisation works for the other fields. However, the Ansatz for $\mathcal{B}_{\mu\,MN}$ cannot be directly deduced from the expressions in 2009.03347 (I would imagine deducing it from an Ansatz for the constrained two-forms in $D=4$, but this is not explcit there either). In particular, it should be explained why it is acceptable that their fluctuations are not independent from those of the standard vector fields. Were the equations of motion used to derive this result? Is the last line of (3.2) only motivated "a posteriori" by consistency of the linearised equations of motion and explicit results in later sections?

Also, the following points are very minor but worth pointing out

2) At the beginning of the Introduction the author refers to past literature on KK spectroscopy on coset spaces. The sentence gives the impression that the techniques derived here apply for more general manifolds, but in fact globally generalsed parallelisable spaces are known to always be (topologically) homogeneous (see 0807.4527 section 5.3).

3) Related to the previous point, it seems to me that the $\rm G_{max}$ group introduced around equation (3.3) is really the group that has a transitive action on the coset space, or perhaps its compact subgroup? I would suggest to clarify this point in view of future applications.

---

## Round 1 · Referee Report · Anonymous (Referee 2) · 2021-4-22

Strengths

  1. Clear presentation throughout, with self contained summary of relevant background at the beginning of the paper.
  2. Results are technically sound and important in the context of both the AdS/CFT correspondence and in applications of exceptional field theory.

Weaknesses

No significant weaknesses

Report

This paper concerns the computation of the Kaluza-Klein spectrum for AdS3 vacua, using the more efficient methodology from exceptional field theory rather than explicit harmonic analysis. The paper builds on a number of works by the Lyon group and does not introduce particularly new techniques. Nonetheless the results obtained in this work are important in the context of holographic correspondences and also in exploring how the exceptional field theory programme can be developed. The most interesting analysis in this work relates to the case in which group theory alone does not fix the spectrum i.e. the AdS3 x S3 x S3 x S1 background, a duality that has been very fruitful for study over many years. The paper is very clearly written and presented and could be published with no changes.

---

## Round 2 · Author Response

I would like to thank the referees for their comments and remarks. The response to the referees is given with the list of changes made in the improved version of the article.

---

## Round 2 · List of Changes

1) This is a very good point. Indeed, I considered fluctuations for ${\cal B}_{\mu\,MN}$ that depend on the ones of ${\cal A}_{\mu}{}^{MN}$ to ensure the consistency of the linearized equations of motion. The discussion has been modified after Eq. (3.3).

2) The techniques do indeed apply to topologically coset spaces, I thank the referee for pointing this out. However, the new technique derived in the paper applies to both manifolds with small or large isometry groups, while the standard harmonic analysis is suited for backgrounds with large isometry groups only. The discussion has been clarified in the introduction.

3) I clarified before Eq. (3.4) that Gmax is compact and has a transitive action on the coset space.

Typos have also been corrected in Eq. (2.16), (3.16), (3.17), (3.24) and (3.25), and Ref. [33] has been added in footnote 12.

---

## Editorial Decision

published